

# A bright idea—metabarcoding arthropods from light fixtures

Vasco Elbrecht[1,2], Angie Lindner[3], Laura Manerus[2] and Dirk Steinke[2,4]

[1] Department of Environmental Systems Science Institute of Biogeochemistry and Pollutant Dynamics (IBP), ETH Zurich, Zurich, Switzerland
[2] Centre for Biodiversity Genomics, University of Guelph, Guelph, Ontario, Canada
[3] Centre for Biodiversity Monitoring, Zoological Research Museum Alexander Koenig, Bonn, Germany
[4] Department of Integrative Biology, University of Guelph, Guelph, Ontario, Canada

## ABSTRACT

Arthropod communities in buildings have not been extensively studied, although humans have always shared their homes with them. In this study we explored if arthropod DNA can be retrieved and metabarcoded from indoor environments through the collection of dead specimens in light fixtures to better understand what shapes arthropod diversity in our homes. Insects were collected from 45 light fixtures at the Centre for Biodiversity Genomics (CBG, Guelph, Canada), and by community scientists at 12 different residential homes in Southern Ontario. The CBG ground floor of the CBG showed the greatest arthropod diversity, especially in light fixtures that were continuously illuminated. The community scientist samples varied strongly by light fixture type, lightbulb used, time passed since lamp was last cleaned, and specimen size. In all cases, the majority of OTUs was not shared between samples even within the same building. This study demonstrates that light fixtures might be a useful resource to determine arthropod diversity in our homes, but individual samples are likely not representative of the full diversity.

## INTRODUCTION

Humans have always been sharing their living spaces with a large variety of organisms ranging from microbes to vertebrates. Arthropods, as the most diverse group of animals (*Mora et al., 2011*), are a key component of this biodiversity at our homes (*Sattler et al., 2011*), and even in seemingly unhospitable environments such as modern urban settlements some of them thrive and even continue to provide a number of ecosystem services (*Prather et al., 2013*). It is however, poorly understood how arthropod diversity is influenced by increased urbanization (*Mata et al., 2017*) or by the drastic alterations of ecosystems that are currently fragmenting populations and putting natural habitats at risk (*Steffen et al., 2015*). What we do know is that biodiversity in our homes is richer than it was once thought (*Bertone et al., 2016*) and while most of our small cohabitants are harmless, others can be vectors for allergies or diseases. Some can damage food, clothing, or building structure, thereby indirectly affecting human well-being (*Bertone et al., 2016*). In order to mitigate any potential detrimental effects and to better manage risks posed

Corresponding author
Dirk Steinke, dsteinke@uoguelph.ca

by some arthropod species, it is critical to understand which species have found their way into our homes (*Schoelitsz, Meerburg & Takken, 2019*).

Only very few studies have investigated the diversity of arthropods in homes. One of the most extensive of them surveyed 50 homes in North Carolina, USA by collecting insects from all visible surfaces in all rooms of free-standing homes (*Bertone et al., 2016*). Results showed that insect assemblages in homes are influenced by several factors, such as the fauna living around the house and the form and availability of access points to homes. Other studies related community composition to the presence of carpets (*Leong et al., 2017*) as well as to socio-economic factors (*Leong et al., 2016*). Interestingly, factors such as the level of tidiness or the use of pesticides, as well as pet ownership seemed to have no significant influence on community composition (*Leong et al., 2017*). However, authors often note that one potential shortcoming of their studies could have been insufficient taxonomic identification, which was often restricted to family level (*Bertone et al., 2016*). Identification of many arthropod groups is often challenging, due to lack of identification keys, availability of taxonomic experts, damaged specimens or life stages which do not allow for a reliable species level identification. Especially when sample sizes are thousands of specimens, identification higher than family or even order level becomes a substantial time constraint.

DNA-based methods of species delimitation such as DNA barcoding (*Hebert et al., 2003*) can substantially speed up the identification process and reliably provide species level identities. Especially when using DNA barcoding for bulk samples (DNA metabarcoding), the species composition of entire communities can be rapidly determined (*Ritter et al., 2019*; *Steinke et al., 2021a*, *2021b*). Studies used metabarcoding of house dust to detect arthropods (*Madden et al., 2016*) and pollen (*Craine et al., 2017*) in hundreds of samples collected across the USA. With high sample size and increased taxonomic resolution they were able to show that location of the home, presence of pets and the presence of a basement had significant effects on arthropod community diversity (*Madden et al., 2016*). Capturing such information from house dust using DNA-based methods has the advantage that no taxonomic expertise is needed for collecting or identifying contents of the samples. Consequently, sample collection can be done by community scientists.

For this study we explored if arthropod DNA can be retrieved from indoor environments through the collection of dead specimens in light fixtures. As many arthropods are attracted to light sources (*Shimoda & Honda, 2013*) they often accumulate in light fixtures for many years. Such samples might be ideal for metabarcoding, as specimens are dry, which means the DNA is likely fairly well preserved. We tested the feasibility of a metabarcoding approach to explore arthropod communities collected in light fixtures of 12 residential homes in Ontario and did some more standardized sampling within the Centre for Biodiversity Genomics (CBG, Guelph, Canada) where we were able to collect samples from the same lamp type across the entire building to test for reproducibility and representativeness of samples.
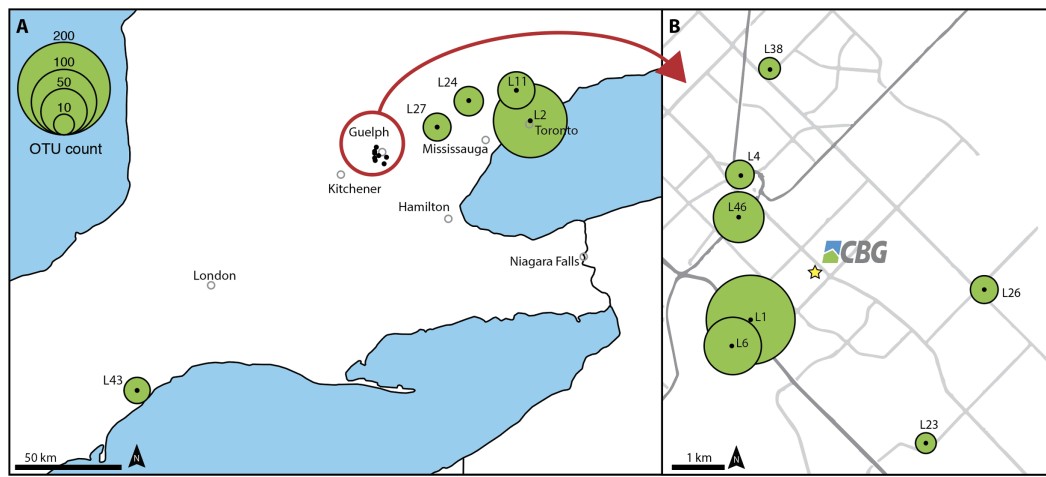

**Figure 1 Location of the 12 insect samples collected by community scientists.** Map (A) shows the samples collected in Ontario, Canada, with section (B) providing a detailed overview of samples collected in the City of Guelph. The Centre for Biodiversity Genomics (CBG) where additional samples were collected is marked with a yellow star. The area of each green circle corresponds to the number of arthropod OTUs detected in the respective sample.

## MATERIALS AND METHODS

### Sample collection

Arthropods were collected by community scientists in early 2019 at 12 locations in Southern Ontario, Canada. They used gloves to remove the remains from individual lamps and to transfer them into a 50 ml falcon tube. All participants were asked to fill out a survey to provide details about sampling location, room type, personal habits as well as an estimate as to when the light fixture was last cleaned (see Table S1 for details). In addition, in April 2019, we collected samples from 45 light fixtures within the Centre for Biodiversity Genomics (CBG) building (Fig. 1B). Light fixtures in basement, ground floor and first floor are identical and equipped with the same type of incandescent light bulbs, thereby ensuring comparable sampling conditions across floors. The light fixtures measure 58 × 58 cm with an area of 20 × 58 cm in which specimens can accumulate (see Fig. S2). Eighteen of the 45 lights are automatically turned off every night for 10 h, as part of the building's energy saving system (highlighted in Fig. S1). All light fixtures were last cleaned in August 2015. Samples were collected directly into 2 ml reaction tubes or 20 ml IKA Ultraturax tubes, with two or ten steel beads (Ø 4 mm) added for sample grinding, depending on the amount of specimens (Fig. S3). Community science samples were handled identically.

### DNA extraction and metabarcoding

We followed a scalable metabarcoding laboratory workflow that uses 96 well microplates (*Elbrecht & Steinke, 2019*). The plate included additional 22 freshwater macroinvertebrate samples of a different project, 12 negative controls and nine extraction blanks. Eight of the latter were filled with extracted DNA of another project (see Table S2 for plate map). Samples with higher biomass were ground in IKA Ultraturax tubes for 30 min at 4,000

rpm, while samples in 2 ml tubes were ground for 90 s at 28 Hz. Approximately 20 mg of ground tissue was used for DNA extraction using a DNeasy 96 Blood and Tissue Kit (Qiagen, Hilden, Germany). To reduce the risk of cross contamination tissue was aliquoted and lysed in individual 1.5 ml tubes. During incubation at 56 °C for 3 h, tubes were gently inverted 5 times every hour to ensure mixing of tissue and lysis buffer. After cooling and subsequent brief centrifugation, 410 µL of AL buffer was added to the lysate and the tube immediately shaken. Subsequently, the lysate was centrifuged at 16,000*g* for 1 min to precipitate cuticula fragments. The supernatant was transferred to the column extraction plate. The remainder of the DNA extraction process was carried out following manufacturer's protocols. Extraction success was visualized on a 1% agarose gel (Fig. S4A).

Metabarcoding was carried out using a two-step fusion primer PCR protocol (*Steinke et al., 2021a*). During the first PCR step, a 421 bp region of Cytochrome c oxidase subunit I (COI) was amplified using the BF2 + BR2 primer set (*Elbrecht & Leese, 2017*). PCR reactions were carried out in 25 µL reaction volume, with 0.5 µL DNA, 0.2 µM of each primer and 12.5 µL PCR Multiplex Plus buffer (Qiagen, Hilden, Germany). We used a Veriti thermocycler (Thermo Fisher Scientific, Waltham, MA, USA) with the following cycling conditions: initial denaturation at 95 °C for 5 min; 25 cycles of: 30 s at 95 °C, 30 sat 50 °C and 50 sat 72 °C; and a final extension of 5 min at 72 °C. PCR success was checked on a 1% agarose gel (Fig. S4B). One µL of PCR product was used as template for the second PCR, where Illumina sequencing adapters were added using individually tagged fusion primers (see Fig. S1 in *Elbrecht & Steinke (2019)*). Tagging combinations are available in Table S2). The same PCR conditions as in the first PCR step were used, but the reaction volume was increased to 35 µL, cycle numbers were reduced to 20 and the extension time was increased to 2 min. PCR success was checked on a 1% agarose gel (Fig. S4C). PCR products were purified and normalized using SequalPrep Normalization Plates (Thermo Fisher Scientific, Waltham, MA, USA; *Harris et al. (2010)*) following manufacturer protocols. Normalization success was checked on a 1% agarose gel (Fig. S4D). Ten µL of each normalized sample were pooled and the resulting library cleaned using left sided size selection with 0.76x SPRIselect (Beckman Coulter, Brea, CA, USA), to remove primer dimers from the negative controls. Sequencing was carried out by the Genomics Facility at the University of Guelph using a 600 cycle Illumina MiSeq Reagent Kit v3 and 5% PhiX spike in. Both indexing read steps were skipped, as inline tags were used. The read length of read one was increased to 316 bp, while keeping read two at 300 bp to increase the number of reads that can be successfully merged.

## Bioinformatic processing

Raw sequence data were quality checked using FastQC v0.11.8, and processed using the JAMP pipeline v0.69 (github.com/VascoElbrecht/JAMP). Scripts for data processing are available in Scripts S1. Samples were demultiplexed using JAMP, and then paired and merged using Usearch v11.0.667 with fastq_pctid=75 (*Edgar, 2010*). Due to parallel sequencing of forward and reverse direction (*Elbrecht & Leese, 2015*), we generated reverse complement sequences when needed using Usearch. Primer sequences were removed from

each sample using Cutadapt v1.18 with default settings (*Martin, 2011*), retaining only sequences where primers were successfully trimmed at both ends. Cutadapt was also used to remove sequences below 411 bp and above 431 bp length. Sequences with poor quality were removed using an expected error value of 1 (*Edgar & Flyvbjerg, 2015*) as implemented in Usearch. The filtered reads of each sample were de-replicated and singletons removed, before pooling all reads for OTU clustering using Usesarch cluster_otus with 97% similarity threshold. Duplicated reads of each sample including singletons were mapped back against generated OTUs using usearch_global, to generate an OTU table. The maximum read count for each OTU found in the 12 negative controls was multiplied by two, and subtracted from all other samples to reduce the effects of low abundance tag switching and cross-contamination (*Elbrecht & Steinke, 2019*). OTUs below 0.01% abundance in at least one sample were removed. Taxonomic assignment of OTUs used the JAMP Bold_web_hack script and the BOLD reference database (www.boldsystems.org; *Ratnasingham & Hebert, 2007*) similar to earlier studies (e.g. *Steinke et al., 2021a*). For most analysis carried out in R v3.5.1, relative read counts were used, and only reads above 0.01% abundance considered. All scripts for statistical analysis are available in Scripts S1. For statistical analysis only Arthropod OTUs were considered. In order to look at potential correlations between estimated specimen size and read counts we obtained sizes for species found mostly from *bugguide* which also served as source for habitat information (https://bugguide.net/node/view/15740, accessed May 20, 2021). Some sizes were estimated from images obtained from BOLD (accessed May 20, 2021, if those contained length references). Further details are available in Table S3.

## RESULTS

We were able to extract high quality DNA for most samples, and obtained strong bands for all 57 samples after the second PCR step (Fig. S3A). Illumina sequencing produced 21.555.865 reads. Raw data is available on NCBI SRA under the accession number SRP200520.

The community scientist samples varied strongly by light fixture type, lightbulb used, time passed since lamp was last cleaned, and specimen size (Table S1). OTU richness recovered by metabarcoding varied (Fig. 1) with limited overlap between samples. Overall, 82% of the OTUs were present in only one of the 12 samples (Fig. 2). The two samples with the highest OTU counts, one from a townhouse in Guelph and the other one from an apartment at the 20th floor in a Toronto high-rise building, only showed 9.9% overlap. Taxonomically, samples were dominated by Diptera, Coleoptera, Hymenoptera, Hemiptera, and Lepidoptera. The community science samples also contained a few low abundant non-target OTUs matching Ascomycota, Basidiomycota, *Wolbachia* and a few mammals (*Bos taurus*, *Canis lupus*, Table S3). The sample with *Canis lupus* matches was taken in a household with two dogs.

Samples collected at the CBG showed some OTU overlap between different floors (Fig. 3), but very limited overlap between individual samples (Fig. 2B), 57.1% of the arthropod OTUs were only found in one of the 45 samples. There is a significant

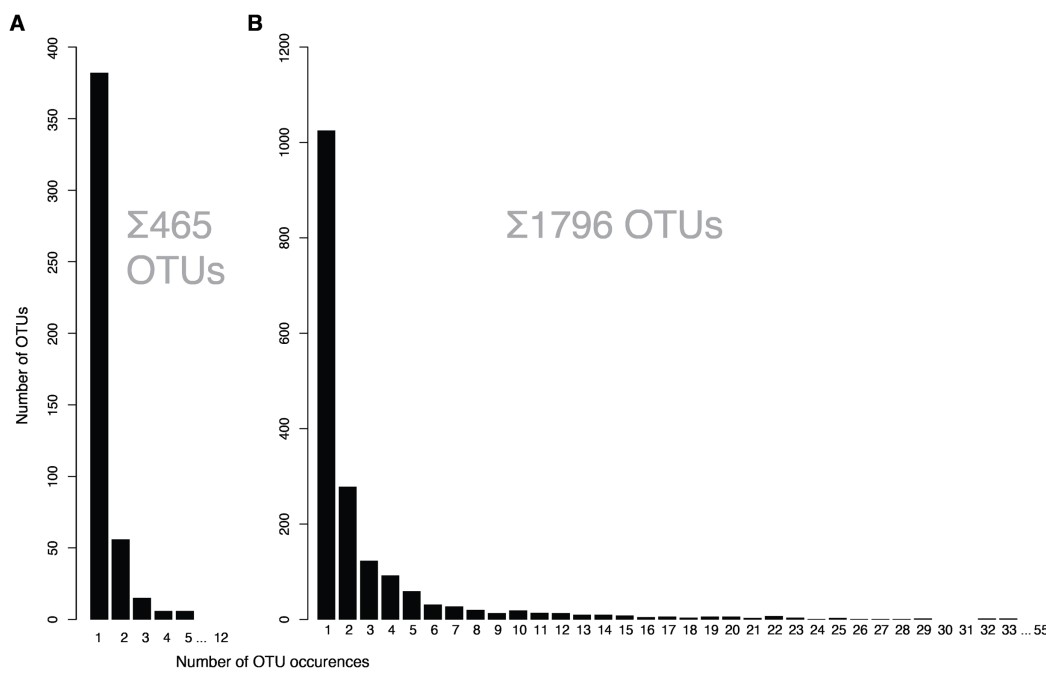

**Figure 2 Histogram showing arthropod OTU sharing between samples collected in homes (A) and at the CBG (B).**

correlation between estimated specimen size and read counts (Fig. 4). Large specimens were usually limited to only a few samples, smaller specimens with high read counts were present in more samples, while small specimens with low total abundance were limited to few samples. Samples from the basement were dominated by OTUs assigned to *Clogmia* sp. which made up an average of 66.6% (SD = 18.2) of reads for the basement samples (Fig. S5). There was also a significant difference between OTU abundance in light fixtures that are turned off overnight and the ones that stay on permanently (Fig S6, *t*-test: $p < 0.0001$).

A barcode reference dataset of 13,149 Arthropod specimens collected outside of the CBG from May to October 2013 is available on BOLD under the accession DS-MTBIO (subset of dataset from *Hebert et al. (2016)*). Around 1/3 (35.96%) of the OTUs collected inside the CBG did match to specimens collected outside with 98% similarity or higher (only considering arthropod OTUs matching the full BOLD database with 98%).

## DISCUSSION

Our study demonstrates that metabarcoding of dead arthropods from light fixtures can a feasible approach to determine their diversity inside buildings. As sampling can be carried out by tenants and home owners, studies of arthropod biodiversity in our homes can become much less intrusive and more engaging through community science projects (*Bertone et al., 2016*).

As expected, most of the taxa detected in our light fixture samples were flying arthropods, with only a few exceptions (Spiders, *Porcellionides pruinosus, Scutigera*

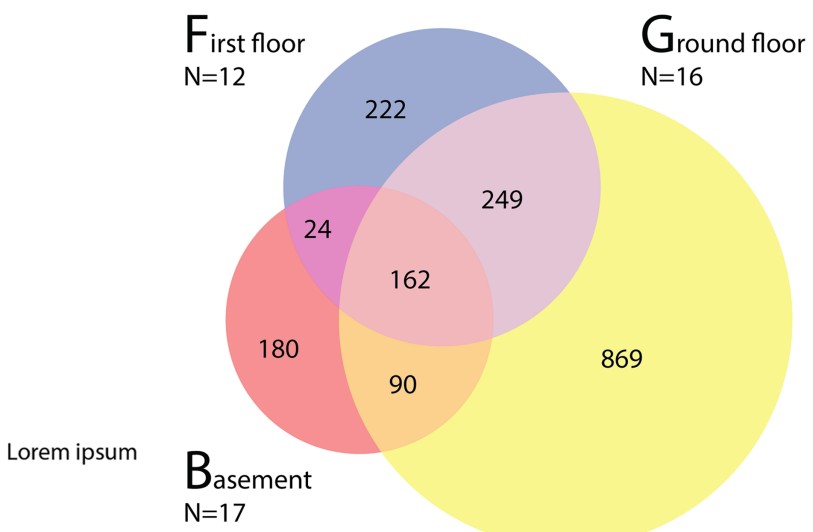

**Figure 3 Arthropod OTU sharing between CBG basement, ground and first floor.**

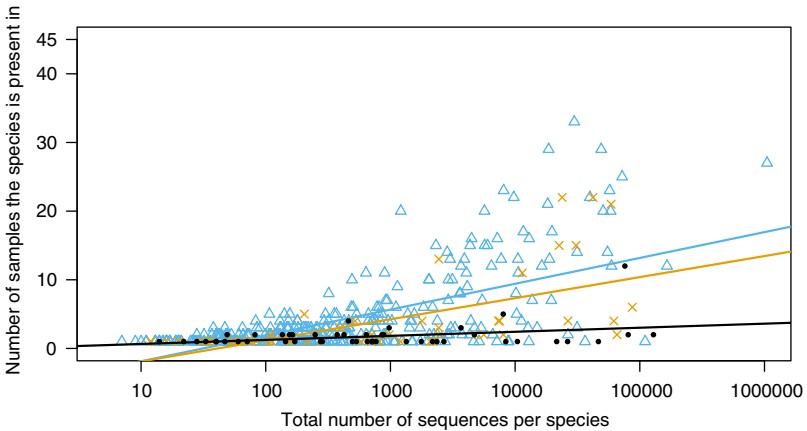

**Figure 4 Species shared between samples collected at the CGG, dependent on amount of reads.**

*coleoptrata*) and trace amounts of non-arthropod reads (bacteria, molds, Clitellata). The taxonomic composition was very similar to what can be found when using Malaise traps (*Karlsson et al., 2005*) with Diptera usually making up at least half of the samples followed by Hymenoptera and Hemiptera (*deWaard et al., 2019*, *Steinke et al., 2021a*, *2021b*). The detection of Clitellata might stem from gut content of scavenging arthropods or low levels of laboratory cross contamination (other scientists in the laboratory are working on earthworms, OTU_1164 was present in 2 negative controls each with two reads). Similarly, *Asellus aquaticus* was found in a few negative controls at a read depth of 62 or less. This aquatic organism likely represents cross-contamination from other projects. Such low abundant cross-contaminations highlight the need for multiple negative

controls, as well as the inclusion of extraction replicates and the manual validation of both the OTU table and taxonomic assignments (*Elbrecht & Steinke, 2019*). However, they didn't pose a major issue for this study. Nonspecific amplification of mold and bacteria could become a problem when degenerate primers are used in the presence of large amounts of non-target DNA (*Hajibabaei et al., 2019*; *Pereira-da-Conceicoa et al., 2019*). However, DNA was extracted from dry bulk samples and mostly intact as verified after DNA extraction using gel electrophoresis. A majority of the reads recovered were arthropod reads, giving further evidence that DNA degradation was not a concern. Another expected non-target organism is *Wolbachia* because many arthropods are infected with these bacteria (*Hilgenboecker et al., 2008*; *Smith et al., 2021*). We also detected mammal DNA from cattle (*Bos taurus*) and wolf/dog (*Canis lupus*). Sources of mammalian DNA could be gut content from biting flies such as mosquitoes (*Reeves et al., 2018*; *Lynggard et al., 2019*) or from scavenging arthropods feeding on meat in the household, and hair of pets. The wolf/dog detection is most likely an example of the latter as one of the two homes with the detection reported dogs as pets.

The overlap between the individual community science samples was very limited. Most taxa detected were unique to the individual homes. As the overall sample size of 12 is fairly low the limited overlap could be caused by incomplete sampling of a highly diverse insect community. Other factors such as the light fixture type, the last time the fixtures were cleaned, the fauna living around the home, and the presence of carpet (*Leong et al., 2017*; *Madden et al., 2016*) likely have substantial influence on the amount and types of arthropods that can be recovered. For a better understanding of these factors, we recommend the use of a more detailed survey for participants and more extensive sampling of more rooms as well as areas outside the home. *Leong et al. (2017)* demonstrated that the insect community found inside is influenced by the physical features of a room, such as the number of windows and whether the room has carpeting. Most insects we detected were flying insects, thus carpeting will likely have no major influence on the community composition in light fixtures, but the number of doors and windows likely will. It also would be important to measure the biomass of the collected arthropods, to find out if low abundance of taxa detected is the result of low biomass or of limited community diversity. The sample L02 collected in a Toronto high-rise building is particularly interesting, as it was one of the community science samples with the highest diversity, despite being located on the 20th floor. Previous studies (*Bertone et al., 2016*; *Leong et al., 2017*; *Madden et al., 2016*) were limited to smaller homes, but high-rise buildings might show a diversity gradient of insect communities based on elevation. Assuming inflow of insects by wind drift is more limited at higher floor levels (*Hardy & Milne, 1938*), human habits such as frequent ventilation by keeping windows open, as well as insect size and susceptibility to passive transport might play a bigger role than on ground level. However, the current data set with its limited sample size does not allow further exploration of these ideas.

By sampling 45 light fixtures at the CBG, we were able to gain some insights into insect diversity in a larger office building, an environment that has not yet been studied. Such buildings usually contain less plants and food, but are frequented by more people (staff and visitors) which in turn might influence arthropod communities.

Within the CBD, we found only limited overlap between individual samples, but as all light fixtures sampled were of the same type we analysed them in different combinations. Various patterns emerged when averaging results by floor or when grouping them according to lights being turned off overnight or not. For example, bathrooms in the basement seem to be more frequented as they are more secluded and quiet, which might have led to a drain fly (*Clogmia* sp.) infestation. Additionally, while many taxa are only present in a single light fixture, there is more overlap between samples by floor level. Insect diversity is highest at the ground level which is in contrast to earlier results (*Leong et al., 2017*) where the basement showed the highest richness. However, the high abundance of *Clogmia* in our basement samples could have reduced detection of other taxa by metabarcoding (*Elbrecht, Peinert & Leese, 2017*). Nevertheless, the ground level is more frequented and provides more opportunities for arthropods to enter the building when doors are opened. As windows are equipped with screens to stop arthropods from entering doors likely represent the primary pathway for insects to enter the building. This opens the possibility of developing further strategies to reduce the influx of arthropods further by e.g. the installation of air curtains.

We also found significantly more taxa in light fixtures that are kept on overnight which is not surprising as many arthropod taxa are attracted by light (*Shimoda & Honda, 2013*). However, many of those taxa were also found in light fixtures that are turned off at night likely because they still attract animals throughout the day.

We found substantial size effects, e.g. specimens larger than 10 mm showed a high read abundance but were limited to only a few samples. In addition, many of the smaller specimens were limited to only a few light fixtures. In contrast, small taxa with high read counts were usually present in several samples. These stochastic effects, leading to limited overlap between samples can be explained by the general paucity of taxa in this ecosystem with limited resources or by a deliberate avoidance of buildings. Especially for large taxa resource limitations might lead to a limitation of abundance, as it takes more resources to maintain a larger body size. Species with smaller body size can reach higher abundance, which also was the case in our study as taxa that co-occurred in ten or more light fixtures were usually <10 mm. We hypothesize that these effects might also have an influence on insect sampling outside of homes, as when most insect taxa are rare, it will be fairly random if they end up in malaise or pitfall traps (*Steinke et al., 2021a*, *2021b*). Such subsampling effects are especially pronounced for rare taxa, while smaller more common species are more likely to be captured.

## CONCLUSIONS

We were able to show that light fixtures in buildings can act as passive terrestrial arthropod traps over longer periods of time. Bulk specimens collected from light fixtures yield high quality DNA that in most cases can be easily metabarcoded because samples are already dry which facilitates homogenisation. While these samples contain useful biological signals, such as past infestation with pests, or effects of floor level, the signal between light fixtures seems to be stochastic. The 45 identical light fixtures probed at CBG captured a lot of taxa present in one sample only. This might indicate that the diversity of

arthropods is large and a single light fixture will only capture a limited amount of diversity. This methodological limitation, as well as the heterogeneity of homes, light fixtures and surrounding environments, impedes the collection of statistically meaningful data from such projects unless sample sizes are increased considerably. Nevertheless, light fixtures can be a good source to survey arthropod communities in homes, representing another tool to better understand diversity in residential and commercial buildings.

## ACKNOWLEDGEMENTS

This work represents a contribution to the 'Food From Thought' research program of the University of Guelph. We would like to thank all participating community scientists for contributing samples.

### Funding

This work was supported by the Canada First Research Excellence Fund. The funders had no role in study design, data collection and analysis, decision to publish, or preparation of the manuscript.

### Grant Disclosures

The following grant information was disclosed by the authors:
Canada First Research Excellence Fund.

### Competing Interests

The authors declare that they have no competing interests.

### Author Contributions

- Vasco Elbrecht conceived and designed the experiments, performed the experiments, analyzed the data, prepared figures and/or tables, authored or reviewed drafts of the paper, and approved the final draft.
- Angie Lindner analyzed the data, authored or reviewed drafts of the paper, and approved the final draft.
- Laura Manerus performed the experiments, authored or reviewed drafts of the paper, and approved the final draft.
- Dirk Steinke conceived and designed the experiments, authored or reviewed drafts of the paper, and approved the final draft.

### Data Availability

The raw sequence data is available in the NCBI SRA: SRP200520.

Reference sequences for taxa collected with malaise trapping outside of the CGB are available on BOLD: DS-MTBIO.

## Supplemental Information

Supplemental information for this article can be found online at http://dx.doi.org/10.7717/peerj.11841#supplemental-information.

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
