# Peer review of "A bright idea—metabarcoding arthropods from light fixtures"

_PeerJ, doi:10.7717/peerj.11841_

## Round 0.1 · original submission · Minor Revisions

· Academic Editor

Minor Revisions

As you'll notice, both reviewers were enthusiastic about your study. However, they identified a few minor issues that should be adjusted before the paper can be accepted, particularly Reviewer 2.

·

Basic reporting

The article 'A bright idea? Metabarcoding arthropods from light fixtures' is an interesting study that proposes a new tool for identifying dead insects collected from light fixtures in urban environments. The text is well written with a correct English easy to understand. The authors have well-posed the problem, which to my knowledge and in its context is very relevant. The conclusions correspond well to the results obtained. Overall, the figures are well presented, except for figure 4 which should be redone.

Experimental design

The authors have shown clarity in their approach with a level of detail that facilitates its reproducibility. However, the approach has some limitations in capturing the full diversity of species, favoring more flying arthropods and smaller specimens. However, the authors have well discussed its limitations in the discussion. All additional data were accessible and well-informed.

Validity of the findings

The results from the study are robust enough to validate the feasibility of metabarcoding to identify dead species collected in light fixtures.

Additional comments

No general comments for the author as I find the article in its current state to be publishable.

Some minor comments:

Line 52: Remove the comma.
Fig. S5: Give the meaning of the gray color bars

Reviewer 2 ·

Basic reporting

I consider the manuscript to be very high quality, well written, and with an appropriate coverage of the literature. The article is well structured with good quality figures and tables.
The raw data is already in suitable repositories, and all the necessary information and metadata, that would allow external validation, is available.

Experimental design

The manuscript is comprised of original primary research that falls within the scope of the journal. The study aims to explore a novel source of biodiversity data and the research question is clear and well defined.

The methods are robust, current, and described with sufficient detail to replicate. The inclusion of the scripts for data analysis further enhances replicability & reproducibility. However, there are a few instances where details are missing that would otherwise allow the reader to better understand the rationale behind certain processes.

1. Why have the author's increased the sequencing length of Read 1 (line 142)? Whilst it's clear that these cycles are available to be used, as no external indexes are involved, the benefit of doing so for a 421bp amplicon is not. Does this lead to an increased number of reads being successfully merged?

2. What is the rationale for deducting 2x the numbers of reads found within the negative controls from each of the other samples? I understand the need to account for background contamination (e.g. errors caused by index hopping) but would be concerned about removing rare taxa. Especially as you highlighted that the methods employed are biased towards the recovery of larger and more abundant taxa.

Validity of the findings

Between the figures for the manuscript and the ample supplementary files, the authors have provided all underlying data from the study. However, as an aside, it would be nice if some of the taxonomic information had made it in to the main body of the manuscript. While the aim of they study is to assess a novel sampling strategy, the underlying purpose is to monitor biodiversity and this seems to have been lost in relegating that information to then end of a table in the supplementary materials.

The conclusions are well stated and supported by the results. The authors haven been open about the limitations of the study and the strength of conclusions that can subsequently be drawn.

Additional comments

This is a nice, well-thought-out study exploiting an overlooked source of urban invertebrate samples that could easily become a large-scale citizen science project.

---

## Round 0.2 · accepted · Accept

· Academic Editor

Accept

I believe that all of the remaining issues have been properly addressed.